# Prevalence of specific micronutrient deficiencies in urban school going children and adolescence of India: A multicenter cross-sectional study

**Shally Awasthi** [1] *, **Divas Kumar**[1], **Abbas Ali Mahdi**[2], **Girdhar G. Agarwal**[3], **Anuj Kumar Pandey**[1], **Hina Parveen**[2], **Shweta Singh**[4], **Rajiv Awasthi**[5], **Harsh Pande**[1], **Anish T. S.**[6], **B. N. Mahanta**[7], **C. M. Singh**[8], **Joseph L. Mathew**[9], **Mohammad Kaleem Ahmad**[2], **Kuldeep Singh**[10], **Mushtaq A. Bhat**[11], **Somashekar A. R.**[12], **Sonali Kar**[13], **Suma Nair**[14]

1 Department of Pediatrics, King George's Medical University, Lucknow, Uttar Pradesh, India, 2 Department of Biochemistry, King George's Medical University, Lucknow, Uttar Pradesh, India, 3 Department of Statistics, University of Lucknow, Lucknow, Uttar Pradesh, India, 4 Department of Psychiatry, King George's Medical University, Lucknow, Uttar Pradesh, India, 5 Prarthana Diabetic Care Centre, Lucknow, Uttar Pradesh, India, 6 Department of Community Medicine, Government Medical College, Thiruvananthapuram, Kerela, India, 7 Department of Medicine, Assam Medical College, Dibrugarh, Assam, India, 8 Department of Community & Family Medicine, All India Institute of Medical Sciences, Patna, Bihar, India, 9 Department of Pediatric Medicine, Post Graduate Institute of Medical Sciences, Chandigarh, India, 10 Department of Pediatrics, All India Institute of Medical Sciences, Jodhpur, Rajasthan, India, 11 Department of Pediatrics, Sher-i-Kashmir Institute of Medical Sciences, Srinagar, Jammu & Kashmir, India, 12 Department of Pediatrics, M. S. Ramaiah Institute of Medical Sciences, Bangalore, Karnataka, India, 13 Department of Community Medicine, Kalinga Institute of Medical Sciences, Bhubaneswar, Orissa, India, 14 Department of Community Medicine, Kasturba Medical College, Manipal, Karnataka, India

* shally07@gmail.com

## Abstract

### Introduction

Childhood and adolescence require adequate amount of micronutrients for normal growth and development. The primary objective of study was to assess the prevalence of deficiencies of Vitamins (Vitamin A, 25 Hydroxy Vitamin D, Vitamin B12 and Folate) and minerals (Calcium, Zinc, Selenium and Iron), among urban school going children aged 6–11 and 12–16 years in ten cities of India. Secondary objective was to find the association between micronutrient deficiencies with sociodemographic and anthropometric indicators.

### Methods

A multi-center cross-sectional study was conducted across India. Participants in the age groups of 6 to 11 years (group 1) and 12 to 16 years (group 2) were selected from randomly chosen schools from each center. Data on socio economic status, anthropometric measures was collected. Blood samples were collected for biochemical analysis of micronutrients. Point estimates and 95% confidence intervals was used to assess the prevalence of deficiencies. Associations were observed using chi square, student t test and ANOVA test.

**Data Availability Statement:** All relevant data are within the paper.

**Funding:** This work was supported by a grant from Hindustan Unilever Limited (Grant Number: 212332). Funding supports all study related expenses including manuscripts processing fees. Funding source was not involved in study design, implementation, collection and interpretation of data and in writing of the manuscript.

**Competing interests:** The authors declare that they have no competing interests.

## Results

From April 2019 to February 2020, 2428 participants (1235 in group 1 and 1193 group 2) were recruited from 60 schools across ten cites. The prevalence of calcium and iron deficiency was 59.9% and 49.4% respectively. 25 Hydroxy Vitamin D deficiency was seen in 39.7% and vitamin B12 in 33.4% of subjects. Folate, Selenium and Zinc were deficient in 22.2%, 10.4% and 6.8% of subjects respectively. Vitamin A deficiency least (1.6%). Anemia was prevalent in 17.6% subjects and was more common among females.

## Conclusion

One or more micronutrient deficiencies are found in almost one half of school going children in urban area. Hence efforts must be made to combat these on priority.

## Trial registration number

CTRI/2019/02/017783.

## Introduction

Vitamins and minerals, termed together as micronutrients, although required in small quantities, are vital for the normal growth, development and functioning of human body. These are essential throughout the life but the period of childhood and adolescence is more important, as it is marked by rapid growth and development. Review of literature suggests that deficiency of minerals (calcium, iron, selenium and zinc) and Vitamins (Vitamin A, 25 Hydroxy Vitamin D, Vitamin B12 and Folate) during childhood and adolescence have negative impact on general health, growth, neuropsychological behavior, cognitive and motor development, intelligence quotient, attention, learning, memory, language ability and educational achievement [1–9]. Iron, folate or vitamin B12 deficiency results in anaemia which has a negative impact on work capacity, intellectual performance, and child cognitive development [1]. Vitamin A plays a critical role in eye health and immune function [1]. Zinc deficiency can interfere with multiple organ systems including brain development and cognition, particularly when it occurs during a time of rapid growth and development [2].

Micronutrient deficiencies affect an estimated two billion people, or almost one-third of the world's population [10]. In India, around 0.5 per cent of total deaths in 2016 were contributed by nutritional deficiencies [11].

National surveys in India, like National Family Health Survey 4 (2014–15) [12] reported high prevalence of amaemia among children aged 6 to 59 months, women aged 15 to 49 years ($\geq 50\%$) and 22.7% among men aged 15 to 49 years. The National Nutrition Monitoring Bureau [13] reported 67% to 78% prevalence of anaemia among preschool children, adolescent girls, pregnant and lactating women in rural areas. These surveys including Coverage Evaluation Survey (2009) [14], Annual Health Survey (2012–13) [15], District Level Household Survey (2012–13) [16] had evaluated the nutritional status based on anthropometric measurements, dietary intake and presence of anemia [17, 18]. These surveys did not assess the micronutrient status, except National Nutrition Monitoring Bureau (2012), which reported prevalence of Vitamin A deficiency as 0.2%, on basis of Bitot's spots. Many surveys to assess the burden of various micronutrient deficiencies were limited to state or district [17, 19–28]. Hence, there is scarcity of a nationally representative data on micronutrient status of children and adolescents.

The aim of this current study is to fill this knowledge gap by collecting data through a well-planned, large-scale study using standardized methodologies to assess the prevalence of various micronutrient deficiencies, among urban school going children. The primary objective of study was to assess the prevalence of deficiencies of Vitamins (Vitamin A, 25 Hydroxy Vitamin D, Vitamin B12 and Folate) and minerals (Calcium, Zinc, Selenium and Iron), among urban school going children aged 6–11 and 12–16 years in ten cities of India. Our secondary objective was to find the association between micronutrient deficiencies with sociodemographic and anthropometric indicators.

## Methodology

### Ethical approval

The study was approved by the institutional ethics committee of each site. The study was registered prospectively with Clinical Trial Registry of India (registration number CTRI/2019/02/017783).

### Study design

This is a multi-center cross-sectional study conducted in ten major district across India. In each district, about 240 participants from randomly chosen government and private schools, located within urban limits of the city were recruited. The detailed study protocol is published elsewhere [29].

### Study setting

Districts of Bangalore, Bhubaneswar, Chandigarh, Dibrugarh, Jodhpur, Lucknow, Patna, Srinagar, Thiruvananthapuram and Udupi (Manipal) were selected to collect nationally representative data (Fig 1). King George's Medical University (KGMU), Lucknow was the central coordinating unit (CCU) for the study.

These ten districts have a total urban population of 21.2 million, which is 5.6% of country's total urban population. Urban population aged 6 to 16 years was 4 million in these cities, which accounts to 5.2% of country's population in this group [30].

### Sample size computation

The sample size was calculated using folate deficiency as this gave the maximum sample. Assuming the prevalence of folate deficiency in India as 30.7% [31], precision (d) of 2% and level of significance (α) 0.05, the calculated sample size [32] was 2044 participants. After taking, attrition rate of 10% sample size inflate to 2400 participants. This sample size was equally divided among the ten districts.

### Selection of participants

Participants were selected by using two-stage sampling technique. In the first stage, schools were selected, and in the second stage participants were recruited from the selected schools. Each study district provided a list of schools imparting co-education to children between 6 to 16 years of age and located within the urban limits of city. From this list, six schools were selected using simple random sampling by GGA, having at least one to a maximum of three private schools. The government aided schools were considered as government schools only. There were more government schools than private schools. Selection of schools was done in similar ratio.

Principals of recruited schools were met to obtain voluntary written informed consent. With the help of an identified coordinating school teacher, a gender-wise list of students between 6 to 11 (group 1) and 12 to 16 years (group 2) of age was prepared. From each of these

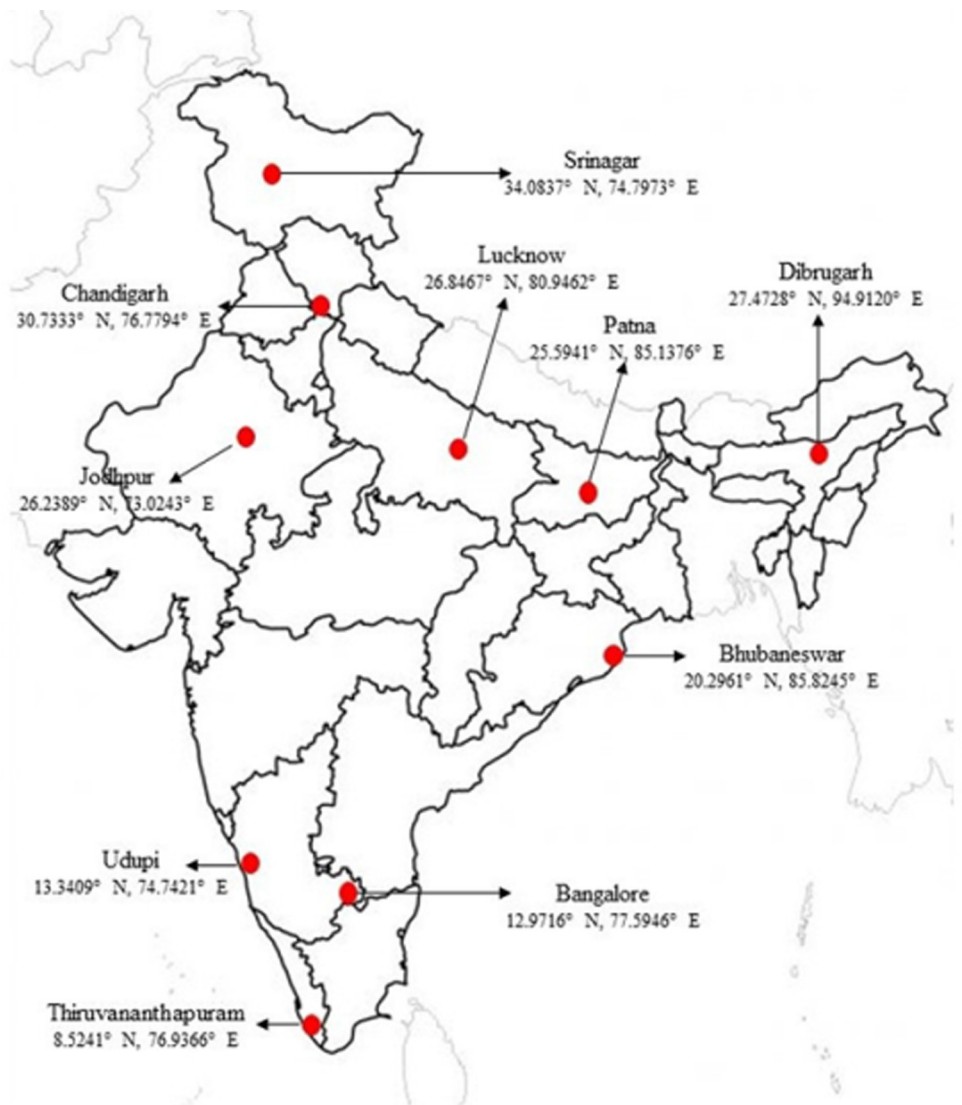

**Fig 1. Study districts and their geo-coordinates.**

lists, fifteen students who were apparently healthy and residing within five kilometers of radius from school were selected by random draw. They were invited to participate into the study. Out of these, first ten participants whose parents had provided written informed consent were included in the study. Rest was kept as back-up in case of exclusions. Written assent was obtained from all participants who were 8 years or above of age. After obtaining consent, participants were assessed for body mass index (BMI). Those having BMI less than 12.5 were excluded from the study and their parents were informed and advised to seek medical consultation.

## Training of study team

Data collectors were trained at CCU Lucknow by project investigators and co-investigators, in four batches from March to July 2019. They were trained on study protocol procedures, data collection instruments, data entry tools and standard operating procedures.

## Data collection

The data was collected from April 2019 and March 2021. There was delay in data collection due to COVID-19 pandemic. Since, the school calendar of different states of India varies from another owing to their specific geography, culture and climate, time frame of data collection across the districts also varied accordingly.

**Demographic and socioeconomic data.** Demographic and socioeconomic details were recorded by interviewing participants and their primary care giver. Revised Kuppuswamy's socioeconomic scale [33] was used to assess socioeconomic status.

**Anthropometric measurements.** Anthropometry was done by qualified and trained nutritionists. The height was measured to the nearest 0.1 centimeter using Seca 213 Mobile Stadiometer (Seca, Hamburg, Deutschland). Two measurements of height were recorded for each participant. If the difference between the two height measurements was greater than 5 mm, then a second set of two height measurements was taken to obtain more precise values.

Weight was measured to the nearest 0.1 kilogram using portable Seca 803 weighing scale (Seca, Hamburg, Deutschland). The unit was standardized by calibrating it to zero before each measurement.

**Blood collection, storage and transport.** Blood sampling was done at school by trained phlebotomists, during early school hours. Venous blood sample of 6 ml (4 ml in clot activator and 2 ml in EDTA vial) was collected using vacuum-tube systems, preferably from cubital vein. All aseptic precautions were taken. Students were kept under observation for 15 minutes after sample collection.

During transportation of blood sample from school to study sites, temperature was maintained between 2˚C to 8˚C, using pre frozen gel packs in ice box. Within two hours, these samples were centrifuged at 1500 rpm for 10 minutes at 4˚C to separate plasma, serum and packed cells, at the study site. Plasma and serum were stored below -20˚C in trace element-free cyro tubes and packed cells between 2˚C to 8˚C, at the study sites. Samples from study districts to CCU were transported in two batches of 120 each, maintaining temperature below -20˚C for plasma and serum and between 2˚C to 8˚C for packed cells, by professional agencies having expertise in handling and shipment of bio-medical samples. Samples were prevented from exposure to light during the whole process.

**Laboratory methods.** Analysis of blood samples was done at Department of Biochemistry, KGMU. Quantitative determination of zinc and selenium in the serum was done by Inductively Coupled Plasma-Optical Emission Spectrometer (ICP-OES, Optima 8000, Perkin Elmer, USA) [34]. The level of vitamin D, Vitamin B12 and folate were analyzed in serum by Chemiluminescent Microparticle Immunoassay (CMIA) technology with flexible assay protocol provided by kit ARCHITECT 25-OH VITAMIN D, ARCHITECT B12 and ARCHITECT Folate (Abbot Diagnostics, Iceland) respectively [35–37]. Serum calcium and iron were measured by fully automatic analyzer by Selectra PRO M. Calcium and Iron kits were provided by calcium arsenazo III colorimetric Labkit, Delhi, India and ELITGroup empowering IVD, USA, respectively [38, 39]. Vitamin A was estimated in serum using ELISA (Enzyme Linked Immunosorbent Assay) method according to the manufacturer's recommendation (USCN Wuhan USCN Business Co., Ltd. cat# CED051Ge). The unlabeled retinol in the standards and samples undergo competitive inhibition with biotin labeled retinol, provided in ELISA plate. Following a series of steps, the concentration retinol is estimated inversely with the amount of HRP conjugated at 450nm in spectrophotometer.

Internal and external quality control of analysis was maintained by running controls along with the samples and by participation in external quality assurance programs.

**Table 1. Deficiency cut off levels of micronutrients and hemoglobin.**

| | Age group | Cut off levels | Reference |
|---|---|---|---|
| Calcium | 6 to 16 years (male & female) | <10 mg/dl | [42, 43] |
| Iron | | <70 µg/dl | [44, 45] |
| Selenium | | < 5.5 µg/dl | [46] |
| Zinc | < 10 years (male & female) | <65 µg/dl | [47] |
| | ≥ 10 years (male) | <70 µg/dl | |
| | ≥ 10 years (female) | <66 µg/dl | |
| Vitamin A | 6 to 16 years (male & female) | <20.0 µg/dl | [48] |
| Marginal Vitamin A | | 20.0 µg/dl ≤ Vitamin A level <30.0 µg/dl | [49] |
| 25 Hydroxy Vitamin D | | <12 ng/ml | [18] |
| Folate | | <3 ng/ml | [50] |
| Vitamin B12 | | <203 pg/ml | [18] |
| Mild-Anemia (Hemoglobin levels) | 6 to 11 years (male & female) | 11.0 to 11.4 g/dl | [41] |
| | 12 to 14 years (male) | 11.0 to 11.9 g/dl | |
| | 12 to 16 years (female) | | |
| | 15 to 16 years (male) | 11.0 to 12.9 g/dl | |
| Moderate-Anemia (Hemoglobin levels) | 6 to 16 years (male & female) | 8.0 to 10.9 | |
| Severe-Anemia (Hemoglobin levels) | 6 to 16 years (male & female) | < 8.0 | |

### Data management and statistical analysis

Double data entry was done in MS excel. Data was matched electronically and discrepancies were rectified by referring to the source documents. For computing stunting (height for age) and BMI categories, we used WHO Anthro Survey Analyzer [40]. Micronutrient deficiencies were estimated using the cut off levels as given in Table 1. Anemia was defined on basis of hemoglobin was estimated using the WHO defined age and gender specific cut-off [41] (Table 1).

Point estimates and 95% confidence intervals of proportions of different micronutrient deficiencies were evaluated. Chi square test was used for comparison of proportions. Mean and standard deviation of micronutrient levels (continuous variables) were computed and compared using student t- test. ANOVA test was used for comparing more than two variables. P < 0.05 was used as statistically significant level.

To use the nested structure of children within school, multilevel model was used in addition to individual level model to find the association of important factors (sociodemographic profile like age, gender, family type, mother's education, socioeconomic class and anthropometric indicators like severe thinness/thinness, overweight, obese/severe obese, stunting) with outcome variable (micronutrient deficiency).

## Results

Recruitment of subjects in the study was done from April 2019 to Feb 2020. In this duration 2428 participants, aged 6–11 years (Group 1) and 12–16 years (Group 2), were recruited into the study from 60 schools [37 (61.6%) government and 23 (38.3%) private] across 10 participating cities of India. There were almost similar proportion of males and females. The sociodemographic profile of study participants is given in Table 2.

### Micronutrient status

The prevalence of specific micronutrient deficiencies is given in Table 3. Most common mineral deficiency was of calcium (59.9%, CI 58.0%-62.0%), followed by iron (49.4%, CI 47.0% - 52.0%), whereas zinc (6.8%, CI 6.0% - 8.0%) was least deficient.

**Table 2. Sociodemographic and anthropometric characteristics of study participants.**

| Age group | Group 1 (6–11 years) | Group 2 (12–16 years) |
|---|---|---|
| | (m/n, %) | |
| *School Type (Row %)* | | |
| Government | 784/1524, 51.4 | 740/1524, 48.6 |
| Private | 451/904, 49.9 | 453/904, 50.1 |
| *Type of family (Row %)* | | |
| Joint | 415/743, 55.9 | 328/743, 44.1 |
| Nuclear | 820/1682, 48.8 | 862/1682, 51.2 |
| *Mother's education (Row %)* | | |
| Illiterate (no formal education) | 214/434, 49.3 | 220/434, 50.7 |
| Class 1 to 9 | 439/850, 51.6 | 411/850, 48.4 |
| Class 10 to 12 | 411/804, 51.1 | 393/804, 48.9 |
| Graduate or above | 168/328, 51.2 | 160/328, 48.8 |
| *Socio-economic class (Row %)* | | |
| Upper | 17/36, 47.2 | 19/36, 52.8 |
| Upper Middle | 176/366, 48.1 | 190/366, 51.9 |
| Lower middle | 523/1001, 52.2 | 478/1001, 407.8 |
| Upper lower | 464/928, 50.0 | 464/928, 50.0 |
| Lower | 55/94, 58.5 | 39/94, 41.5 |
| *Anthropometric Indicators (Column %)* | | |
| *Height for age* | | |
| Stunted (HAZ < –2 SD) | 158/1235, 12.8 | 204/1193, 17.1 |
| *BMI for age (Column %)* | | |
| Severe thinness / Thinness | 157/1235, 12.7 | 202/1193, 16.9 |
| Normal | 862/1235, 69.8 | 820/1193,68.7 |
| Over-weight | 139/1235, 11.3 | 124/1193, 10.4 |
| Obese / Severe Obese | 77/1235, 6.2 | 47/1193, 3.9 |

25 Hydroxy Vitamin D, vitamin B12 and folate were all deficient in more than one quarter of the subjects, whereas vitamin A was least deficient (< 2%). We found 39.7% (CI 38.0% - 42.0%) of study population was deficient in serum Vitamin D. Prevalence of Vitamin B12 deficiency was 33.4% (CI 31.0% - 35.0%), and that of folate was 22.2% (CI 21.0% - 24.0%). Vitamin A deficiency was 1.6% (CI 1.0% - 2.0%) and marginal vitamin A deficiency was 6.1% (CI 5.1% - 7.1%).

Further, comparing the proportion of deficiency, in gender by age group, we found statistically significant difference in 25 Hydroxy Vitamin D ($p<0.001$) in group 1 and in iron ($p<0.001$), 25 Hydroxy Vitamin D ($<0.001$) and vitamin B12 ($p = 0.004$) in group 2.

In multilevel analysis, the school level effect was found to be statistically non-significant (i.e., variance component due to schools was not statistically significant from zero). Therefore, associations were observed only at individual level using traditional multiple logistic regression.

Males were at higher odds of deficiency of Vitamin B12 (crude odd ratio 1.24, 95% CI, 1.04–1.47, $p = 0.02$) and zinc (crude odd ratio 1.49, 95% CI, 1.05–2.12, $p<0.03$) than females. Whereas, females were at higher odds of deficiency of iron (crude odd ratio 1.57, 95% CI, 1.33–1.85, $p<0.001$) and vitamin D (crude odd ratio 2.28, 95% CI, 1.92–2.71, $p<0.001$) than males. Comparing by age group, older age group (12–16 years) was having higher odds of deficiency of folate (crude odd ratio 1.68, 95% CI, 1.37–2.05, $p<0.001$) and vitamin B12 deficiency (crude odd ratio 2.09, 95% CI, 1.75–2.45, $p<0.001$) than younger age group (6–11 years).

**Table 3. Specific micronutrient deficiencies by age group and by gender.**

| Micronutrient deficiency | Prevalence of deficiency | Group 1 | | Group 2 | |
|---|---|---|---|---|---|
| | m/n, % (95% CI) | Male | Female | Male | Female |
| | | m/n, % Deficient | | m, % Deficient | |
| Calcium | 1358/2267, 59.9 | 347/683, 50.8 | 336/683, 49.2 | 322/675, 47.7 | 353/675, 52.3 |
| | (58.0–62.0) | | | | |
| Iron | 1119/2265, 49.4 | 283/573, 49.4 | 290/573, 50.6 | 214/546, 39.2 | 332/546, 60.8 |
| | (47.0–52.0) | | | | |
| Zinc | 138/2026, 6.8 | 42/67, 62.7 | 25/67, 37.3 | 40/71, 56.3 | 31/71, 43.7 |
| | (6.0–8.0) | | | | |
| Selenium | 208/2003, 10.4 | 50/107, 46.7 | 57/107, 53.3 | 48/101, 47.5 | 53/101, 52.5 |
| | (9.0–12.0) | | | | |
| Vitamin A | 37/2250, 1.6 | 11/21, 52.4 | 10/21, 47.6 | 8/16, 50.0 | 8/16, 50.0 |
| | (1.0–2.0) | | | | |
| 25 Hydroxy Vitamin D | 900/2268, 39.7 | 168/441, 38.1 | 273/441, 61.9 | 173/459, 37.7 | 286/459, 62.3 |
| | (38.0–42.0) | | | | |
| Folate | 505/2276, 22.2 | 104/205, 50.7 | 101/205, 49.3 | 140/300, 46.7 | 160/300, 53.3 |
| | (21.0–24.0) | | | | |
| Vitamin B12 | 759/2275, 33.4 | 150/292, 51.4 | 142/292, 48.6 | 257/467, 55.0 | 210/467, 45.0 |
| | (31.0–35.0) | | | | |

Based on hemoglobin concentration, 17.6% (425/2410, 95% CI, 16.1–19.2) participants were anemic, of which 63.3% (269/425) were mild, 34.4% (146/425) moderate and 2.3% (10/425) severe anemic. We observed higher prevalence of anemia in females compared to males (crude odd ratio 2.4, 95% CI, 1.9–3.0, p<0.001). A statistically significant difference was observed between the mean hemoglobin between the two age groups (12.6±1.0 for group 1 and 12.9±1.4 for group 2, p<0.001) and by gender (13.1±1.2 for males and 12.4±1.2 for females, p<0.001).

Comparing the proportion of deficiency with BMI categories, we found statistically significant difference for calcium, zinc and selenium as shown in Table 4. Since the prevalence of vitamin A deficiency was low (1.6%), so we have not compared it with BMI categories.

## Discussion

Micronutrient malnutrition is a persistent problem among the low- and middle-income countries [51]. Globally, an estimated 29% of preschool children have vitamin A deficiency, 18%

**Table 4. Comparison of micronutrient deficiency in participants across BMI categories.**

| Micronutrient | Overall deficiency n/N (%) | BMI | | | | |
|---|---|---|---|---|---|---|
| | | Severely Thin / Thin | Normal | Over weight | Obese / Severely Obese | p value |
| | | m/n, column % | | | | |
| Calcium | 1358/2267, (59.9) | 213/341, 62.5 | 966/1578, 61.2 | 125/236, 53.0 | 54/112, 48.2 | **0.004** |
| Iron | 1119/2265, (49.4) | 164/341, 48.1 | 777/1578, 49.2 | 110/234, 47.0 | 68/112, 60.7 | 0.089 |
| Zinc | 138/2026, (6.8) | 10/307, 3.3 | 82/1407, 5.8 | 29/209, 13.9 | 17/103, 16.5 | **<0.001** |
| Selenium | 208/2003, (10.4) | 14/303, 4.6 | 129/1391, 9.3 | 37/207, 17.9 | 28/102, 27.5 | **<0.001** |
| 25 Hydroxy Vitamin D | 900/2268, (39.7) | 123/341, 36.1 | 626/1582, 39.6 | 96/233, 41.2 | 55/112, 49.1 | 0.10 |
| Folate | 505/2276, (22.3) | 69/343, 20.1 | 349/1587, 22.0 | 56/234, 23.9 | 31/112, 27.7 | 0.35 |
| Vitamin B12 | 759/2275, (33.4) | 124/343, 36.2 | 538/1586, 33.9 | 70/234, 29.9 | 27/112, 24.1 | 0.72 |

have anemia and 17% of the population are at risk of inadequate zinc intake [52]. India is also facing the challenge of micronutrient malnutrition as reported by data from Comprehensive national nutrition survey (CNNS) [18] which was published in 2019, at the time when the current study was operationalized. CNNS lacks the data about calcium and selenium status of population. The current study was conducted in 2019–21 to find the prevalence of micronutrient deficiency as estimated by serum levels, in school children aged 6–11 and 12–16 years, studying in urban schools of ten cities of India. Almost one fourth of children had deficiency of one or more vitamin among vitamin D, B12 and folate. Vitamin A deficiency was found in less than 2% of them. More than half the population had deficiency of calcium and iron whereas zinc and selenium deficiencies were found in about 10%. The findings of this study provide an insight into the current status of micronutrient malnutrition in urban school going children aged 6 to 16 years. Despite the fact that several nutritional programs are operational [53], the prevalence of multiple micronutrients deficiency is still high in the country [54, 55].

Calcium and Vitamin D are complimentary nutrients for bone health. Childhood and adolescence are crucial period for skeletal mineralization. Nutritional trends from independence (1947) to recent years (2016) shows that there is a consistent gap between the production and consumption of dietary calcium in Indian dietary pattern [56]. The current study shows that 59.9% (95% CI, 58.0–62.0) of study participants had calcium levels below 10 mg/dl and 39.7% (95% CI, 38.0–42.0) had Vitamin D levels below 12 ng/ml. This is consistent with the finding that about 80–85% of Indian population is suffering from various degrees of vitamin D deficiency (VDD) [56]. Various studies, in contrast to our study, have reported prevalence of VDD ranging from 88.6% to 93.0% [19–21], although these were small studies conducted in a particular geography. This high prevalence may possibly be attributed to their considering 50 nmol/L (30 ng/ml) as the cut-off level of deficiency. We used a lower cut-off level as suggested by CNNS [18], which reported 18.2% (95% CI, 16.5–20.1) and 23.9% (95% CI, 21.9–26.0) prevalence of VDD, in 5–9 years and 10–19 years age group respectively. Our finding suggest prevalence of VDD is higher than this, possibly because our study was limited to urban settings only. Further, our findings shows that VDD is more prevalent in elder age groups and among females, which is supported by data from CNNS. The common reasons for the deficiency in India include increased indoor life style, pollution, changing food habits, application of sunscreens and cultural practices [57]. These deficiencies result in skeletal deformities, increased risk of infectious disease such as tuberculosis, upper respiratory tract infections of viral origin, auto immune disorders, type 2 diabetes mellitus and obesity, which adds burden to the public health infrastructure [57]. With this background of unsatisfactory levels of twin nutrient deficiency of calcium and vitamin D in India, strategies like population-based education, targeted supplementation and fortification may be implemented [56]. Food fortification may be one of the best strategies, as it has already shown its potential with the success of salt iodization program in India.

Anemia is one of the major public health problems in India as well as globally, affecting nearly one-third population [58]. We found that based on hemoglobin concentration, 17.6% participants were anemic. Deficiencies of iron, folate and vitamin B12 are the major cause behind the anemia in India [22]. Our findings suggest that half of the population is having deficient iron levels and one third are having vitamin B12 levels below 203 pg/ml. Vitamin B12 deficiency is more common in older age group than youngers, as complemented by findings of CNNS [18]. Limited nationwide data is available on vitamin B12 deficiency. However, several studies limited to a state, reported varying magnitude of vitamin B12 deficiency in school going children from 7.4% to 72.7% [22–24]. We found that prevalence of folate deficiency is lower than that of vitamin B12. One in every fourth participant was having serum folate levels below 3ng/ml. Although, school based-cross sectional studies conducted in state of

Maharashtra [45] and National Capital Territory (NCT) [25] reported prevalence of folate deficiency to 40.2% and 39.8% respectively, in participants aged 11–18 years. Findings of our study and that of CNNS [18] shows that folate deficiency is more common in older age group than youngers.

Anemia has multi-dimensional effect on child health and development. Iron deficiency has a negative impact on cognition, behavior, and motor skills [5], while chronic anemia may impair growth and cardiac function too [59]. The prevalence of anemia in India had decreased over the years in due to various national programs like National Nutritional Anemia Prophylaxis Programme (1970), National Nutritional Anemia Control Programme (1991) and the Weekly Iron and folate Supplementation Programme (2013). However, a large population is still suffering from anemia, which needs to be addressed through revisiting the current national program [17].

Vitamin A is an essential for regulating many critical functions of human body including vision, epithelial integrity and the expression of several hundred genes [60]. We found that 6.1% (137/2250) participants were having marginal and 1.6% participants are having vitamin A deficiency. Contrary to our findings CNNS reported 17.54% [61] prevalence of vitamin A deficiency. This may possibly be the due to change in regulation notified by Government of India, on mandatory Vitamin A fortification of cooking oil in year 2018 [62]. National Prophylaxis Programme against Nutritional Blindness due to vitamin A deficiency was launched in 1970 and was modified under the National Child Survival and Safe Motherhood Programme in 1994, targeting the pre-school children. Literature review suggests that there had been a marked reduction in prevalence of vitamin A deficiency. Majority of these studies were targeted to pre-school children otherwise limited to a specific geography [17].

Zinc is essential for virtually all processes of human body, including the immune system [63]. Limited data is available on zinc deficiency at national level. Several studies had reported varying degree of zinc deficiency, like 49.4% in adolescents of Delhi [26], 43.8% among children aged 6–60 months [27] and lower prevalence of 18% among children aged six months to five years in Punjab [28]. Our findings suggest a low prevalence of 6.8% (95% CI, 6.0–8.0) compared to that of CNNS [18]. However, it has been predicted that anthropogenic $CO_2$ emissions would increase the prevalence of zinc deficiency by 2.9% by the year 2050, by altering the nutrient profile of staple food crops [64].

Selenium in the form of seleno-proteins, carries out various functions in normal health and metabolism and is essential for the functioning of cardiovascular, endocrine, immune, musculoskeletal, neural, brain and reproductive system [65]. Its deficiency is less commonly studied and reported. This is probably the first study addressing the selenium deficiency at national level. We found one tenth of children (208/2003, 10.4%) are suffering from selenium deficiency. Studies have showed a significant association between selenium status and cure rate for COVID-19 [66–69]. This makes the selenium deficiency a matter of concern for the public health professionals.

In our study, we found statistically significant difference across the BMI categories of participants having calcium deficiency. The proportion of participants with calcium deficiency was higher in severely thin/thin category. A study from Indonesia, reported that level of calcium and vitamin D were not associated with the incidence of stunting [70]. Another study on Ecuadorian children reported that low serum vitamin D levels in underweight and stunted [71].

We also observed statistically significant difference across the various BMI categories in the participants having Zinc and selenium deficiency. The higher proportion of participants with zinc deficiency were found to be overweight, obese / severely obese. Similar to our findings, Gu et al in their meta-analysis found that serum zinc level was significantly lower in obese

children [72]. Many other researchers also reported decrease in zinc levels in overweight, obese and severely obese [73, 74]. Decreased selenium levels are also reported in overweight, obese / severely obese children [75], complementing to our findings (Table 4).

The present study is first of its kind, estimated the prevalence and magnitude of the problem, at country level. However, the current study was limited to the urban population. Due to logistic issues, we equally divided the total sample size across all ten districts, this may have possibly under-represented the districts with higher population densities. A selection bias was also possible as the current study included the participants residing within 5 km radius of the school.

## Conclusion

The present study comprehensively provides information on the status of micronutrient deficiencies in Indian school-going children and adolescents. Almost half of the population was deficient in calcium and iron and one third were having deficiency of 25 hydroxy Vitamin D and Vitamin B12. One or more micronutrient deficiencies were found in almost one half of study population.

## Ethical approval and consent to participate

The study was approved by the Institutional Ethics Committees for MS Ramaiah Medical College and Hospital Bangalore (approval reference number (ARN): MSRMC/EC/AP-02/02-2019), Kalinga Institute of Medical Sciences Bhubaneswar (ARN: KIMS/KIIT/IEC/112/2016), PGIMER Chandigarh (ARN: PGI/IEC/2019/000152), Assam Medical College (ARN: AMC/EC/1430), All India Institute of Medical Sciences Jodhpur (ARN: AIIMS/IEC/2017/765), King Georges Medical University (ARN: 9334/Ethics/R.Cell-16), Kasturba Medical College (ARN: IEC:388/2019), All India Institute of Medical Sciences Patna (ARN: IEC/AIIMS/PAT/153/2017), Sher-i-Kashmir Institute of Medical Sciences (ARN: IEC/SKIMS Protocol # RP 175/2018) and Medical College Thiruvananthapuram (ARN: HEC.No.04/34/2019/MCT). The study was registered prospectively with Clinical Trial Registry of India (registration number CTRI/2019/02/017783). Written informed consent was obtained from parents of all study participants.

## Author Contributions

**Conceptualization:** Shally Awasthi.

**Data curation:** Divas Kumar, Anuj Kumar Pandey.

**Formal analysis:** Girdhar G. Agarwal, Anuj Kumar Pandey.

**Funding acquisition:** Shally Awasthi.

**Investigation:** Shally Awasthi, Divas Kumar, Abbas Ali Mahdi, Hina Parveen, Anish T. S., B. N. Mahanta, C. M. Singh, Joseph L. Mathew, Mohammad Kaleem Ahmad, Kuldeep Singh, Mushtaq A. Bhat, Somashekar A. R., Sonali Kar, Suma Nair.

**Methodology:** Shally Awasthi.

**Project administration:** Shally Awasthi, Divas Kumar.

**Resources:** Shally Awasthi.

**Supervision:** Shally Awasthi, Divas Kumar, Abbas Ali Mahdi, Girdhar G. Agarwal, Anish T. S., B. N. Mahanta, C. M. Singh, Joseph L. Mathew, Mohammad Kaleem Ahmad, Kuldeep Singh, Mushtaq A. Bhat, Somashekar A. R., Sonali Kar, Suma Nair.

**Writing – original draft:** Shally Awasthi, Divas Kumar.

**Writing – review & editing:** Abbas Ali Mahdi, Girdhar G. Agarwal, Anuj Kumar Pandey, Hina Parveen, Shweta Singh, Rajiv Awasthi, Harsh Pande, Anish T. S., B. N. Mahanta, C. M. Singh, Joseph L. Mathew, Mohammad Kaleem Ahmad, Kuldeep Singh, Mushtaq A. Bhat, Somashekar A. R., Sonali Kar, Suma Nair.

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
