## [Decision Letter · Decision Letter 0]

14 Dec 2021

PONE-D-21-31107Prevalence of specific micronutrient deficiencies in urban school going children and adolescence of India : a multicentric cross-sectional studyPLOS ONE

Dear Dr. Awasthi,

Thank you for submitting your manuscript to PLOS ONE. After careful consideration, we feel that it has merit but does not fully meet PLOS ONE’s publication criteria as it currently stands. Therefore, we invite you to submit a revised version of the manuscript that addresses the points raised during the review process.

Specifically  you need to describe the sampling procedure and variables of the study (including those considered for the regression analysis) in better depth. The analysis, including the relevance of the logistic regression analysis, needs to be revisited. The manuscript should be thoroughly edited, the results section has to be properly organized and tables be formatting according to the requirements of PLOS ONE.

We look forward to receiving your revised manuscript.

Kind regards,

Samson Gebremedhin, PhD

Academic Editor

PLOS ONE

Journal Requirements:

Additional Editor Comments (if provided):

Abstract

I don’t think the sentence “In India, around 0.5 per cent of total deaths in 2016 were contributed by nutritional deficiencies.” Is important to justify the study

Background:

Line 67-69: please cited the surveys reported here.

metal >> mineral

The purpose of the study is to fill the scarcity of a nationally representative data on micronutrient deficiencies but the actual study was limited to urban areas. Please, reorganize the purpose of the study accordingly.

Methodology

Please justify why these specific micronutrients were selected for the study.

Line 85: how were these ten major cities selected? How do you assure the representativeness of the sample for the urban population of the country?  

Line 86: Each school contributed equal sample size (about 240). Why it was not possible to take the size of each school into consideration? Or at analysis level, weighted analysis, based on the student size of the schools, should be done.

Sample size was estimated based on folic acid deficiency? The adequacy of the available sample size for measuring the prevalence of other micronutrient deficiencies should also be tested.

Why was the study limited to students living within 5 km radius of the school? Please also discuss the possible implication of this approach (possible selection bias?).

Line 115: BMI less than 12.5%??? BMI in %? Why you did not use BNI-for-age, which is more appropriate for adolescents?

Line 128: please briefly describe the “Kuppuswamy’s socioeconomic scale”

Blood collection and processing: have you done anything to reduce external contamination of blood samples for trace minerals? This is especially important for zinc analysis.

Table 1:  please improve the formatting. Table 1 and 2 can also be merged.

The variables of the study (including those considered for the multilevel analysis) should be exhaustively described and listed before the data collection section.

Line 182-185: what were these “sociodemographic profile and anthropometric indicators” what was the specific dependent variable? Because you have analyzed several micronutrients.

Results

I don’t think table 3 is really important. As you described in the methods section, equal samples of boys and girls have been taken. So the balance observed between boys and girls is not surprising.

In all tables, please only provide the frequency and %, the denominator should rather be provided at the top of the column.

How did you classify socio-demographic status into five?

Table 4: I could not get the point of statistically comparing the sociodemographic and anthropometric characteristics of the two groups. Please remove the statistical testing. The sentence on line 195 and 196 should also be removed.

Please describe the micronutrient status under a separate sub-header.

Contents in table 5 should be concisely described before the table itself.

Line 209:

Reviewers' comments:

Reviewer's Responses to Questions

**Comments to the Author**

1. Is the manuscript technically sound, and do the data support the conclusions?

Reviewer #1: Partly

Reviewer #2: Yes

Reviewer #3: Partly

2. Has the statistical analysis been performed appropriately and rigorously? 

Reviewer #1: Yes

Reviewer #2: Yes

Reviewer #3: No

3. Have the authors made all data underlying the findings in their manuscript fully available?

Reviewer #1: Yes

Reviewer #2: Yes

Reviewer #3: No

4. Is the manuscript presented in an intelligible fashion and written in standard English?

Reviewer #1: Yes

Reviewer #2: Yes

Reviewer #3: Yes

5. Review Comments to the Author

Reviewer #1: General comments

The current manuscript titled “Prevalence of specific micronutrient deficiencies in urban school going children and adolescence of India: a multicentric cross-sectional study” utilizes a multi-centric cross-sectional study to comprehensively assess the prevalence of deficiencies of Vitamins (Vitamin A, 25 Hydroxy Vitamin D, Vitamin B12 and Folic acid) and minerals (Calcium, Zinc, Selenium and Iron), among urban school-going children aged 6-11 and 12-16 years in ten cities of India. This study found that almost half of school-age children in urban areas had one or more micronutrient deficiencies and almost one fourth of children had deficiency of one or more vitamin among vitamin D, B12 and folate. This article does a good job describing the current status of micronutrient malnutrition among school-going children aged 6-16 years in urban India. However, I have several following significant concerns, especially for the major comments.

Major comments:

1. In the Introduction section, the article describes the current status of the prevalence of micronutrient deficiencies worldwide and in India. This is a good point, however, the reason for focusing only on children and adolescents in urban areas is not given later in this section. It is recommended to add the appropriate content.

2. In the Data collection of the methodology section, the article does not list the specific demographic and socioeconomic data collected. Please consider adding related descriptions in this section.

3.The article applied univariate and multivariate analysis to find the associations of important factors (sociodemographic profile and anthropometric indicators) on outcome variable (micronutrient deficiency) in addition to assessing the prevalence of micronutrient deficiencies. However, these were not presented in the abstract and study objectives. Moreover, the presentation of the logistic model is missing in the Methodology section as well as the specific results of the logistic analysis in the Results section. In addition, the discussion section also lacks an in-depth discussion of the relevant content. It is recommended that these contents be supplemented.

4. The discussion section lacks a discussion of the limitations of the study. In addition, the discussion generally revolves around the findings of the prevalence of micronutrient deficiencies. I think this is a good point. However, the discussion is generally illogical and does not provide targeted guidance on how to improve micronutrient deficiencies in children and adolescents in urban India based on the findings of this article. Please consider reorganizing the discussion section to improve this section.

The additional comments:

Abstract

Please add the statistical methods used in the article in the Methodology section of the abstract.

Introduction

1. The Introduction section lacks a specific description of existing relevant studies, which cannot well reflect the shortcomings of the current studies and highlight the research strengths of this study. Please consider adding related descriptions in this introduction.

2. Line 71: Please add the corresponding references to this section.

Methodology

1. Line 89-95: The existing descriptions in the “Study setting” section do not seem to give a good representation of the 10 cities. In the article, “Population aged 6 to 16 years was 7.1 million in these cities.” What percentage of the population aged 6-16 in these ten cities accounts for the total population aged 6 to 16 in the country? Please give the corresponding values.

2. Line 97-98: Please give the formula for calculating the sample size and explain why the folic acid deficiency rate was assumed in the calculation of the sample size.

3. Line 115-116: Please give the reference for the threshold “BMI <12.5%” as an exclusion criterion. In addition, please give the threshold values and references for the four categories of BMI for age in the corresponding section of the methodology.

4. Please be consistent with the term “folate” throughout the article. “Folate” or “folic acid”?

5. Please add the definition of “Stunted” in the Methodology.

Discussion

1. Line 251-252, Line 267: Please add the corresponding references to this section.

2. In the discussion, the authors repeatedly compare their main results with the National Comprehensive Nutrition Survey (CNNS). In my opinion, if these survey data from the CNNS are so important, it is recommended that a brief description of the survey-related elements of the CNNS, such as the survey population and survey time, be given in the article, along with the corresponding references.

Conclusion

Based on the main results of the article, it seems inappropriate to conclude that “with strengthening the existing health and agricultural system and proper implementation of food fortification guidelines, we may expect improvement in the micronutrient status of population in coming years”. Please verify and be cautious in drawing conclusions based on the main findings of the article.

Reviewer #2: Congratulations, very important work! Extremely relevant and of great public health impact. Here are my comments:

INTRODUCTION

Present some data on the national surveys mentioned between lines 66-68.

Between lines 73-78, it seems the informations is repeated.

METHODOLOGY

Why the choice of using folate in the sample size calculation

RESULTS

Is table 3 really necessary? I find it is not adding much to the manuscript

DISCUSSION

Why the difference in type of family and the implications in results was not discussed?

Between lines 230-239, can you present some data comparisons with other countries?

Why results of crossing BMI with micronutrient deficiencies was not discussed?

Most participants were from public schools, did it not implicated in the results?

Please add the study limitations

Reviewer #3: Title and all through the text (e.g. abstract, methods) - I think you mean "multicentre" and not "multicentric".

Abstract line 44-45: I believe the percentages in brackets are the proportions of participants who were deficient of the named vitamins. The statement says that there were deficient in more than one quarter of subjects, but for folate the prevalence of deficiency 22.2% is less than a quarter.

Methodology

- line 82: you should name the approving institutional ethics committee and include the approval reference number and date where available.

- study design - it would be useful to provide more details about how the sites/schools were selected; it is not sufficient to simply say that they were randomly selected. For example, how was the random selection list generated? How did you ensure that the schools selected were representative of the population? etc. This point needs to be further addressed in light of the decision to equally divide the total sample size across all ten sites - this would mean that sites whose catchment areas had higher population densities were under-represented in the sample, since they contributed the same number of individuals to the sample as less population dense areas.

- in selection of participants you indicate that this was a two-stage sample; you need to further describe how the sampling in each stage was conducted: for example, you indicate that the first stage sampling included 1-3 private schools - how did you ensure this, given that you have no control of this in a random sampling; how was the random selection done for the second stage? The method of random selection should be described. Additionally, I think you should replace "site" with "district", as the sites were actually the schools.

- statistical analysis - you state in line 178 that chi-squared tests were used for comparison of proportions, in line 180-181 that t-tests and ANOVA were used to compare means, and in lines 182-185 that logistic regression models were used to explore associations; however, there is nothing in the objectives of the study calling for any comparisons between groups or exploration of associations. If the study aimed to estimate prevalences of micronutrient deficiency, then these analyses were probably pointless. What would be useful to report would be methods for calculating the prevalences that took into account the clustered nature of the data.

Results

- related to the comment above, the column of p-values in Tables 4 and 5 (and any commentary about these) are unnecessary and should be removed, as they don't address any of the objectives of the study (i.e. the study was not set up to investigate whether group 1 and group 2 differed in terms of characteristics and outcomes - if it was then this would be the wrong design for such a study). In any case, this table is supposed to be purely descriptive of the sample included in the analysis. Furthermore the commentary in lines 209-211 do not really address any clear aims of the study. The analyses comparing micronutrient deficiency between males and females, across different age groups, and across BMI are exploratory and should be presented as such.

6. PLOS authors have the option to publish the peer review history of their article (what does this mean?). If published, this will include your full peer review and any attached files.

Reviewer #1: No

Reviewer #2: **Yes: **Giselle Rhaisa do Amaral e Melo

Reviewer #3: No

---

## [Author Response · Author response to Decision Letter 0]

18 Feb 2022

Page no. 12, line no 242-243, it is now written as "Comparing the proportion of deficiency with BMI categories, we found statistically significant difference for calcium, zinc and selenium as shown in table 4."

Data availability statement has been updated.

---

## [Decision Letter · Decision Letter 1]

1 Apr 2022

Prevalence of specific micronutrient deficiencies in urban school going children and adolescence of India : a multicenter cross-sectional study

PONE-D-21-31107R1

Dear Dr. Awasthi,

We’re pleased to inform you that your manuscript has been judged scientifically suitable for publication and will be formally accepted for publication once it meets all outstanding technical requirements.

Kind regards,

Samson Gebremedhin, PhD

Academic Editor

PLOS ONE

Additional Editor Comments (optional):

Reviewers' comments:

Reviewer's Responses to Questions

**Comments to the Author**

1. If the authors have adequately addressed your comments raised in a previous round of review and you feel that this manuscript is now acceptable for publication, you may indicate that here to bypass the “Comments to the Author” section, enter your conflict of interest statement in the “Confidential to Editor” section, and submit your "Accept" recommendation.

Reviewer #2: All comments have been addressed

2. Is the manuscript technically sound, and do the data support the conclusions?

Reviewer #2: Yes

3. Has the statistical analysis been performed appropriately and rigorously? 

Reviewer #2: Yes

4. Have the authors made all data underlying the findings in their manuscript fully available?

Reviewer #2: Yes

5. Is the manuscript presented in an intelligible fashion and written in standard English?

Reviewer #2: Yes

6. Review Comments to the Author

Reviewer #2: (No Response)

7. PLOS authors have the option to publish the peer review history of their article (what does this mean?). If published, this will include your full peer review and any attached files.

Reviewer #2: **Yes: **Giselle Rhaisa do Amaral e Melo

---

## [Editor Report · Acceptance letter]

1 May 2022

PONE-D-21-31107R1 

Prevalence of specific micronutrient deficiencies in urban school going children and adolescence of India: a multicenter cross-sectional study 

Dear Dr. Awasthi:

I'm pleased to inform you that your manuscript has been deemed suitable for publication in PLOS ONE. Congratulations! Your manuscript is now with our production department. 

Kind regards, 

on behalf of

Dr. Samson Gebremedhin 

%CORR_ED_EDITOR_ROLE%

PLOS ONE